# Long-Term Tumor Control Following Targeted Alpha Therapy (TAT) of Low-Grade Gliomas (LGGs): A New Treatment Paradigm?

**DOI:** 10.3390/ijms242115701

**Published:** 2023-10-28

**Authors:** Leszek Krolicki, Jolanta Kunikowska, Dominik Cordier, Nedelina Slavova, Henryk Koziara, Frank Bruchertseifer, Helmut R. Maecke, Alfred Morgenstern, Adrian Merlo

**Affiliations:** 1Nuclear Medicine Department, Medical University of Warsaw, 02-091 Warsaw, Poland; leszek.krolicki@wum.edu.pl (L.K.); jolanta.kunikowska@wum.edu.pl (J.K.); 2Neurosurgery Department, University Hospital Basel, 4031 Basel, Switzerland; dominik.cordier@usb.ch; 3Department of Neurology, Inselspital, University Hospital Bern, 3010 Bern, Switzerland; nedelina.slavova@gmail.com; 4Department of Neurosurgery, Institute of Psychiatry and Neurology, 02-957 Warsaw, Poland; henryk.koziara@gmail.com; 5European Commission, Joint Research Centre (JRC), 76125 Karlsruhe, Germany; frank.bruchertseifer@ec.europa.eu (F.B.); alfred.morgenstern@ec.europa.eu (A.M.); 6Nuclear Medicine and Radiochemistry, University Hospital Basel, 4031 Basel, Switzerland; 7Department of Neurosurgery, Bern and University of Basel, 4001 Basel, Switzerland

**Keywords:** TAT (targeted alpha therapy), Bismuth-213, Actinium-213, brain tumors, low-grade glioma, astrocytoma, oligodendroglioma, substance P

## Abstract

The median survival time has been reported to vary between 5 and 8 years in low-grade (WHO grade 2) astrocytoma, and between 10 and 15 years for grade 2 oligodendroglioma. Targeted alpha therapy (TAT), using the modified peptide vector [^213^Bi]Bi/[^225^Ac]Ac-DOTA-substance P, has been developed to treat glioblastoma (GBM), a prevalent malignant brain tumor. In order to assess the risk of late neurotoxicity, assuming that reduced tumor cell proliferation and invasion should directly translate into good responses in low-grade gliomas (LGGs), a limited number of patients with diffuse invasive astrocytoma (n = 8) and oligodendroglioma (n = 3) were offered TAT. In two oligodendroglioma patients, TAT was applied as a second-line treatment for tumor progression, 10 years after targeted beta therapy using [^90^Y]Y-DOTA-substance P. The radiopharmaceutical was locally injected directly into the tumor via a stereotactic insertion of a capsule–catheter system. The activity used for radiolabeling was 2–2.5 GBq of Bismuth-213 and 17 to 35 MBq of Actinium-225, mostly applied in a single fraction. The recurrence-free survival times were in the range of 2 to 16 years (median 11 years) in low-grade astrocytoma (n = 8), in which TAT was administered following a biopsy or tumor debulking. Regarding oligodendroglioma, the recurrence-free survival time was 24 years in the first case treated, and 4 and 5 years in the two second-line cases. In conclusion, TAT leads to long-term tumor control in the majority of patients with LGG, and recurrence has so far not manifested in patients with low-grade (grade 2) astrocytomas who received TAT as a first-line therapy. We conclude that targeted alpha therapy has the potential to become a new treatment paradigm in LGG.

## 1. Introduction

The common denominator regarding malignancy in grade 2–4 brain-intrinsic glial tumors, despite distinct histological and genetic features, is the relentless tumor cell invasion of normal brain tissue, which eventually leads to a fatal outcome [1]. The variable dynamics of tumor cell proliferation and invasiveness define distinct prognostic subgroups in LGGs and in high-grade gliomas (HGGs), with overall median survival times of 5–8 years in diffuse, invasive, *IDH*-mutant, grade 2 astrocytoma [2]; 10–15 years in grade 2 1p/19q-hemizygous oligodendroglioma [3]; and 12–16 months in glioblastoma (GBM) [4]. Unchecked tumor progression gradually impairs neurological functions, and thus, the social and professional lives of patients with glioma. Since LGG manifests more often in the younger population, the socioeconomic burden is enormous [5,6,7,8,9]. Whether a biopsy is the only initial therapeutic measure selected, or limited or extensive resectioning with neuromonitoring during wake craniotomy, followed by a wait-and-see strategy, adjuvant external beam radiation, or systemic chemotherapy is used instead [7,10], the final outcome is always fatal. In recent years, extensive surgical resectioning has been promoted as it can extend the survival time by reducing the inherent 50% risk of anaplastic transformation [7,11]; however, recurrence still manifests within a 2 cm margin of the primary location [12]. External beam radiotherapy and chemotherapy are sometimes added to the treatment regimen of LGG; however, photon beam radiotherapy can gradually reduce cognitive and memory functions [7,13], and temozolomide has been shown to logarithmically increase the mutational load in glioma cells [14].

The requirements for successful tumor cell targeting are as follows: an effective therapy for malignant gliomas has to eliminate all infiltrating tumor cells, which are spread all over the entire brain, with a decreasing density from the main tumor nodule. The markedly reduced dynamics of tumor cell proliferation and invasion in LGG, as compared with GBM, offer a unique opportunity to assess novel therapeutic approaches [15]. This can be assessed at the cellular level, using a model calculation to illustrate the growth characteristics of glioma cells, irrespective of their grade, to estimate the whole-body cell count of ±10^14^ cells within a 70 kg man [16]. If we assume that a tumor weighs 70 g (mL), it harbors approximately 10^11^ tumor cells (Figure 1) that consist of the tumor nodule and the infiltrating component, which are not readily visible on MRI scans [17]. The resectioning of 99% of a glioma nodule, as shown on an MRI, would still leave behind approximately 10^9^ invasive tumor cells, with this number being logarithmically lower in LGG. Therefore, following an uneventful removal of a nodular brain tumor component, millions to hundreds of millions of invasive glioma cells are still left behind within the adjacent brain area; this area cannot be controlled using unspecific radio-chemotherapy [13,14]. This lack of specificity in standard treatments can be overcome by selecting, for instance, the substance P receptor; this is identical to the neurokinin type 1 receptor (NK1R) [18,19], which is almost exclusively expressed in the CNS compartment in pathological conditions, such as inflammation, trauma, or neoplasia. Moreover, it has limited expression in the interneurons of the spinal pain afferent system [19,20]. We targeted NK1R, a G-protein-coupled receptor that is overexpressed in all grade 2–4 malignant gliomas [19], by constructing a slightly modified DOTA-chelated ligand for glioma therapy [19]. The blood–brain barrier blocks the entry of sufficient quantities of therapeutic compounds into the CNS tumor compartment following an intra-venous or intra-arterial injection [21]. The direct delivery of therapeutic agents into the extracellular space of a brain tumor overcomes this obstacle and allows for drug concentrations to be maximized at the target site.

Intra-tumoral distribution critically depends on the size of the agent used (e.g., monoclonal antibodies (150 KD) or Fab fragments (20 KD) which display slow and limited intra-tumoral distribution following a local injection) [22,23,24,25]. Conversely, small drug-like peptide vectors of less than 2KD, such as modified substance P, rapidly penetrate the targeted area, including the tumor cell-invaded normal brain, within minutes [19,21,26]. The DOTA-peptide vector is labeled with the highly energetic alpha emitters, ^213^Bi or ^225^Ac, which display a mean dose range of ±80 μm (Figure 2). This very favorable dose range limits local neurotoxicity, in contrast to the use of beta emitters, such as ^177^Lut or ^90^Y, at doses ranging from 1 to 5 mm. In addition, the use of alpha particles minimizes the risk of sublethal tumor cell damage due to an ultra-high energy of 5–8.5 MeV [27].

Targeted alpha therapy, using [^213^Bi]Bi/[^225^Ac]Ac-DOTA-substance P, is a novel treatment option for malignant gliomas. Injecting this small (1.8 KD) diffusible radiopharmaceutical directly into the enlarged extracellular tumor space leads to quick and stochastic distribution within the nodular and the peripheral invasive tumor compartment [19,21]. Following specific receptor–ligand binding and internalization [19], the subsequent release of alpha particles leads to irreversible tumor cell death. Phase 1 and 2 trials in recurrent GBM, which assessed the radiopharmaceutical [^213^Bi]Bi/[^225^Ac]Ac-DOTA-substance P, found that a subgroup of patients with GBM, displaying a tumor diameter of <5 cm in combination with a KPS of >70, benefitted the most from TAT [28,29,30]. In this report, we discuss the results obtained using TAT in LGG, which should allow for the estimation of the long-term risk of late brain toxicity following TAT.

## 2. Results

An assessment of the efficacy of TAT in LGG requires a long follow-up interval. The first patient with LGG who underwent TAT was enrolled in June of 1999 (case 1, Table 1), and they received 2 GBq of [Bi^213^]Bi-DOTA-substance P. Due to a lack of experience, a beta emitter was co-injected, followed by temozolomide chemotherapy. Since neither beta irradiation nor chemotherapy cured the malignant gliomas, the long recurrence-free survival time has to be attributed to TAT. Apart from the moderate neurological deficit caused by the initial tumor location, the patient has remained physically and mentally stable over this 24-year period. In astrocytoma cases 2 and 3, a neoadjuvant application of 2 GBq of [^213^Bi]Bi-DOTA-substance P was given, followed by a necrosectomy, as the only treatment modality; this resulted in very long recurrence-free survival times and an excellent neurological and general condition. In case 4, 29 years after the biopsy of an ill-defined lesion in the motor cortex, a diffuse invasive grade 2 astrocytoma manifested in this area, which was treated using TAT, leading to a transient moderate hemiparesis for 6 weeks, followed by a full neurological recovery and a recurrence-free period of 10 years; it finally transformed into a grade IV glioma. In this patient, TAT significantly reduced the drug-resistant seizure frequency from half a dozen hourly focal seizures to 2–3 monthly attacks. In case 5, TAT was administered late in the progressive phase of the diffuse and invasive astrocytoma. In two fractions, 35 MBq of [^225^Ac]Ac-DOTA-substance P was injected into the resectioned cavity, which led to a strong perifocal edema reaction for several months. Over one year, three open biopsies were performed due to a suspicion of malignant transformation; however, only tumor necrosis was detected, and the patient gradually made a good recovery. In cases 6–9 (Figure 3), TAT was used following first-line treatments, such as surgery and standard radiotherapy, thus stabilizing the further course of the disease. Apart from case 5, wherein the patient was of an advanced age, so far, all eight younger LGG patients have not shown any sign of relapse or tumor progression over 2–16 years (median 11 years) following TAT, nor have they shown signs of late neurotoxicity.

*Statistical analysis:* The Bayesian and Frequentist approaches were used to compare survival proportions at 24 months and 60 months between the two study populations (estimation of interest was an odds ratio). The two study populations consisted of (a) 8 low grade astrocytoma grade 2 cases treated with TAT and (b) a large NIH data collection of astrocytoma grade 2 treated between 1999–2010. The probabilities for a relevant beneficial treatment effect for TAT were 75% and 86%, depending on the skeptical prior choice. The median posterior odds ratios at 24 months were as follows: 0.34 with a 90% credible interval between 0.06 and 1.78 (skeptical prior I), and 0.87 (0.62–1.23) (skeptical prior II) when comparing TAT against usual treatments. That is, the chance of dying in the study population treated with TAT was lower by a factor of 0.34 (or 0.87) compared with the patients undertaking usual treatments in the SEER control population (the detailed report can be obtained from the corresponding author).

*Second-line TAT after targeted beta therapy.* Two patients with low-grade oligodendroglioma, who were initially treated with [^90^Y]Y-DOTA-substance P, developed a tumor relapse 11 and 9 years following initial beta therapy, which was treated with TAT. In both patients (cases 10 and 11), the intensification of pre-existing seizures required an increase in anticonvulsant drugs, but eventually, a satisfactory transient stabilization was again achieved for 4 and 5 years, followed by progression and anaplastic transformation.

*Neurotoxicity.* TAT, as a first-line treatment, is well tolerated and leads to a transient perifocal edema reaction that is easily controllable with dexamethasone, in combination with anticonvulsants. In pretreated cases, irrespective of the treatment modality used, the transient inflammatory reaction following TAT can intensify seizure activity (second-line cases 10 and 11 following targeted beta therapy); however, seizure control can also be improved with TAT, as found in case 5. A pre-existing neurological deficit following standard radiotherapy (case 6, hemianopia) or surgery (case 5, mild aphasia; case 9, mild aphasia and slightly reduced fine motricity in the right hand) can become transiently (cases 5 and 9) and permanently (case 6) more pronounced. No differences in toxicity could be detected between Actinium-225 and Bismuth-213, except for a later onset (4–5 days) of the perifocal edema reaction following the injection of Actinium-225.

*Optimal time window for TAT.* Following the logic of the tumor growth curve (Figure 1), TAT has to be administered as early as possible. Ideally, neoadjuvant TAT should precede tumor necrosis resectioning, which should be followed by intra-cavitary and intra-tumoral (residual nodule) TAT, at least in large tumors. Pretreatments, such as standard radiotherapy, chemotherapy, or targeted beta radiotherapy, increase the risk of secondary toxicity, especially in cases with pre-existing neurological deficits, which can be both transient and permanent. It needs to be determined whether dose fractionation minimizes this risk.

QALY: To measure the socioeconomic impact of TAT, the so-called QALY index [31] was determined as the Karnofsky Performance Score (0.1–1) × overall survival time (years). Since LGGs place an increasingly large burden on an individual, on their family, and on society, delaying or even avoiding neurological decay will have a great socioeconomic impact (QALY median 8, range 2.5–22). For comparison, hypothetically prolonging the overall survival time by 5 years in patients with GBM, following TAT administration, with an average KPS of 80, would result in a QALY value of 4.

## 3. Discussion

In this report, we discuss the long-term results obtained using targeted alpha therapy (TAT) to treat low-grade glioma (LGG), an orphan disease that, so far, cannot be cured due to the invasive nature of the glioma cells that invade the entire healthy brain (Figure 4). In phase 1 and 2 studies, this new form of targeted radiotherapy was successfully developed to treat recurrent glioblastomas [28,29,30,32] using radiopharmaceutical [^213^Bi]Bi/[^225^Ac]Ac-DOTA-substance P, which is a linear, small-peptide vector that is repeatedly injected locally into the tumor compartment in GBM. Toxicity was found to be minimal, mainly consisting of a perifocal edema reaction, which is easily controlled with dexamethasone. The maximum tolerated activity per injection was 2 GBq for Bismuth-213 and 20 MBq for Actinium-225 [30]. In LGG, alpha therapy also revealed its potential as a new and effective treatment option. In contrast to GBM, where repetitive injections of the targeting vector are required to achieve temporary tumor control, a single intra-tumoral injection of [^213^Bi]Bi/[^225^Ac]Ac-DOTA-substance P was found to be sufficient to achieve long-lasting tumor control in both grade 2 oligodendrogliomas (n = 1) and diffuse grade 2 infiltrative astrocytomas (n = 8). The finding that a single injection of the radiopharmaceutical was sufficient to achieve long-lasting tumor control in LGG, as compared with only achieving transitory control in GBM using repetitive injections, reflects the vastly different dynamics in the numeric and volumetric expansion of glioma cells into the adjacent normal brain tissue in very aggressive GBM as compared with LGG.

Precise dose calculations require an accurate measurement of the true tumor volume of grade 2–4 malignant gliomas, which are, by definition, composed of a nodular and an invasive component. The nodular tumor component is well defined using MRI. The invasive component, however, is presently not detectable with any method, and it can be massively underestimated, as exemplified in LGG case 7 (Figure 3 and Table 1). In a very young patient with grade 2 astrocytoma, TAT induced a strong edema reaction around the initially small lesion of 2 mL, which was subsequently resected, leading to a resectioned cavity of approximately 5 mL. In the following four years, this small cavity gradually developed into a much larger cavity of ±50 mL via a slow process whereby the perifocal radionecrotic tissue was continuously resorbed; this process finally completely halted after four years. This case allows us to conclude that the initial tumor volume, composed of the nodular and the dense infiltrative component, must have been approximately 50 mL and not 2 mL, as initially assumed on the pre-TAT MRI. In the resorbed zone around the original 2 mL volume, the cell density must have been so high that this entire perifocal area underwent radionecrotic transformation. In addition to the finally established stable tumor margins four years after TAT, the density of the infiltrating glioma cells located beyond these final margins must have been so low that alpha radiation did not leave any detectable traces behind; however, this theoretical assumption needs to be proven in the future (e.g., by using improved imaging tools that will allow for the detection of small cell clusters). Massive underestimations of the true tumor volume, as exemplified in this case, underscore the need for technological progress that will allow for the visualization of the invasive tumor component, not only in LGG, but also in the much more dynamic GBM, which is likely to require more intense and aggressive TAT dosing schedules.

In this pivotal TAT study on LGG that spans across a time period of 1 to 24 years (median of 10 years), incredible long-term recurrence-free survival times were observed following a local injection of [^213^Bi]Bi/[^225^Ac]Ac-DOTA-substance P. In the first case, concerning a low-grade oligodendroglioma (case 1) that exhibited the longest recurrence-free survival time, it is difficult to interpret the TAT effect due to the co-injection of the beta-labeled radiopharmaceutical [^90^Y]Y-DOTA-substance P, which may have enhanced the efficacy of alpha irradiation. In cases 2 to 9, however, neoadjuvant or adjuvant TAT with [^213^Bi]Bi/[^225^Ac]Ac-DOTA-substance P was the only form of intra-tumoral radiotherapy, and with one exception (case 6, Table 1), it was the only form of treatment, in combination with surgery, that led to the long-term recurrence-free tumor control of diffuse invasive grade 2 astrocytoma (median of 11 years; range of 3 to 16 years). Additional photon beam radiotherapy was only applied to one of these eight patients, directly following the second tumor resectioning (case 6, Table 1). The steep dose decay curve of alpha radiotherapy, with an energy deposition within the range of 1–2 tumor cells (Figure 2), prevented the manifestation of late secondary neurological deficits, as distinct from photon beam or beta radiotherapy, both of which can lead to late neurotoxicity. In these first LGG cases treated with alpha therapy, TAT is likely to have completely eradicated the glioma cells in both the main tumor mass following neoadjuvant application and in the invasive zone in most of these cases, without damaging adjacent neurons and astrocytes. Statistical analyses adapted for low case numbers in the orphan disease, using the Bayesian and Frequentists approaches, showed an increased probability that TAT would affect survival times assessed at 24 and 60 months. The effect of TAT in the peripheral invasive zone of a glioma cannot be presently visualized due to the limited resolution of PET screens for [^68^Ga]Ga-DOTA substance P CT-PET. Earlier SPECT studies with [^111^In]In-DOTA substance P/octreotide tracers showed a gradually decreasing distribution pattern of the radiopharmaceutical into the tumor periphery; however, a limitation of these studies is that they used two-dimensional SPECT [21,26].

*Neoadjuvant application.* In LGG, TAT was used prior to (neoadjuvant) and after (adjuvant) surgery, and it was found to work both ways. Some tumors may appear too voluminous, with increased intracranial pressure at the initial presentation, and therefore, patients may undergo an initial debulking surgery. The advantage of the neoadjuvant approach is that it facilitates the distribution of the radiopharmaceutical into the infiltrative component of the tumor by taking advantage of the elevated intra-tumoral pressure gradient and its periphery as a biodistribution “pump” [32].

*Second-line TAT following local beta radiotherapy.* In two oligodendrogliomas initially treated with local beta radiotherapy using [^90^Y]Y-DOTA-substance P, local relapses manifested about 10 years later, and then, they were treated with targeted alpha therapy that helped to stabilize the disease for several more years. Since high-dose local beta radiotherapy is not capable of leading to very long-term tumor control in LGG, and since no relapses have so far been observed in cases initially treated with alpha radiotherapy, it appears safe to conclude that local beta radiotherapy is clearly less efficient than TAT. In addition, it was found to lead to late neurotoxicity due to its 10–50-fold increased range of energy deposition (Figure 2).

*Dose estimates.* Although the volume targeted by alpha irradiation is discontinuous in the periphery beyond the tumor margins visible on MRI, dose calculations can be adapted from simulations of modeled irradiation with [^90^Y]Y-DOTA-substance P [33]. The effective dose deposited at the target site has been calculated to be manifestly higher than that following photon irradiation, in the range of hundreds to >1000 Gray [32]. Such high biologically active energies are required to kill highly resistant glioma cells. This is well known in nuclear medicine due to the successful application of ^131^I to treat thyroid cancer [34].

*Toxicity.* TAT has a very limited toxicity profile within the recommended dose range. In cases pretreated with beta or photon radiotherapy or chemotherapy, mild toxicity can manifest via the perifocal edema reaction following the local injection, and it is not always completely reversible. For example, partial pre-existing hemianopia advanced into a complete deficit following a single injection of 2 GBq [^213^Bi]Bi-DOTA-substance P in a patient (case 6) who had previously undergone two large temporal tumor resections followed by photon beam radiotherapy. It needs to be determined whether such deficits can be avoided via the fractionation of the alpha dose. TAT also has limitations; for instance, it is not suitable as a form of rescue therapy in large and rapidly expanding malignant gliomas that are progressing towards a pre-terminal stage.

*Socioeconomic impact of TAT in LGG.* LGG places a heavy burden on the patient, their family, and their professional work environment (Figure 4). If TAT is started early, neurological deficits can mostly be prevented, and KPS is likely to remain at 100. If long-term tumor control lasting 20 years or longer could be achieved, this would have a tremendous impact on the lives of these patients who would otherwise gradually decline. Regarding socioeconomic modeling, one lost year of life translates into an estimated annual sum of EUR 100′000 (in the US, up to USD 150′000) [35,36]. If a patient with LGG remained recurrence free for 30 years, with a KPS of 90–100, this would total EUR 3′000′000 saved per patient. In a country such as Germany, with approximately 200 new LGG cases each year, this would amount to about EUR 600 million annually. A simpler way to estimate the impact of TAT is to determine the QALY value [31,37,38,39], which is calculated by multiplying KPS (0.1–1) by the years survived since diagnosis. In this LGG study, meaningful QALY values were obtained following TAT, which were between 2.5 and 22 (median of 8, Table 1).

*Conclusions.* Overall, TAT can be considered a novel, promising, therapeutic tool to achieve the long-term control of low-grade oligodendroglioma and astrocytoma (of WHO grade 2). LGGs are considered to be a true orphan disease, comprising approximately 10% of all malignant gliomas. In EU28, with an estimated number of 15′000 new glioma cases annually, the LGG fraction is around 1500 new cases every year. Efforts are now underway to significantly enhance the limited Actinium-225 production, which will provide a basis for the successful TAT treatment of patients suffering from LGG.

## 4. Methods and Materials

The methodologies concerning preparation, radiolabeling, and quality control have been published elsewhere [40]. A total of 11 patients with LGG, treated with TAT, are presented in this report, as follows: 8 low-grade astrocytomas and 3 low-grade oligodendrogliomas (Table 1) in 3 female and 8 male patients, with a median age of 31 years (range 24–64). All tumors had a hemispheric location, 7 on the right side and 4 on the left side. A molecular genetic analysis was performed on 6 of these cases, as genetic tumor profiling only became routinely available around 2010. Histological diagnoses were made by a specialized neuropathologist, and they were reviewed by a second experienced expert. In case 1, 2 GBq of [^213^Bi]Bi-DOTA-substance P was co-injected with 0.8 GBq of [^90^Y]Y-DOTA-substance P. In 2 patients with oligodendroglioma who received 0.8 GBq of [^90^Y]Y-DOTA-substance P, which led to a recurrence-free survival time of 10 and 11 years, and TAT was used as second-line therapy to control relapse. A stereotactic biopsy, followed by neoadjuvant TAT, was used in 3 of the 11 cases, with the subsequent removal of the tumor necrosis. In 8 cases, the resectioning of the nodular tumor component was the first line of treatment. External beam radiotherapy was performed once (case 6), and temozolomide chemotherapy was used twice (cases 1 and 11). Nine patients received a single injection of the radiopharmaceutical, and, in 3 cases, 2–3 injections were administered for insufficient drug distribution (case 9), in order to treat a large progressively growing grade 2 astrocytoma (case 5) and for tri-focal localization in a relapsing oligodendroglioma (case 10). The radiopharmaceutical ^213^Bi was used to radiolabel DOTA-substance P in 8 cases, and it was extracted from a high-dose Actinium-225 generator as described [40]. The Bismuth-213 activity was 2 GBq in 7 cases and 2.5 GBq in case 10, and it was administered as a single injection.

The activity of [^225^Ac]Ac-DOTA-substance P was 17 MBq in case 9, 20 MBq in cases 8 and 10, and 35 MBq in case 5, with the latter being in 2 fractions. The overall survival times were determined by the time between diagnosis (imaging and/or clinical symptoms) and death. The QALY score was calculated by multiplying the Karnofsky Performance Score (KPS) with the number of years of recurrence-free survival time [31].

## Figures and Tables

**Figure 1 ijms-24-15701-f001:**
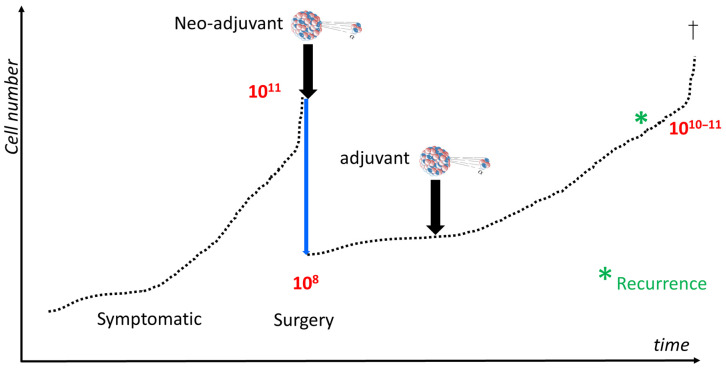
Exponential cell growth in malignant glioma. In a model calculation, a malignant glioma of 70 g has approximately 100 billion tumor cells at the time of diagnosis, assuming a 70 kg man harbors approximately 10^14^ cells in his body. The 99% resection of the tumor mass (blue arrow) still leaves behind hundreds of millions of invasive glioma cells, which gradually reaches the tumor cell number prior to surgery, which is called “recurrence” (denoted as asterisk in green). The earlier TAT sets in, the better the chances are of depositing a sufficient amount of tumoricidal energy within the tumor-infiltrated brain area. († denotes death).

**Figure 2 ijms-24-15701-f002:**
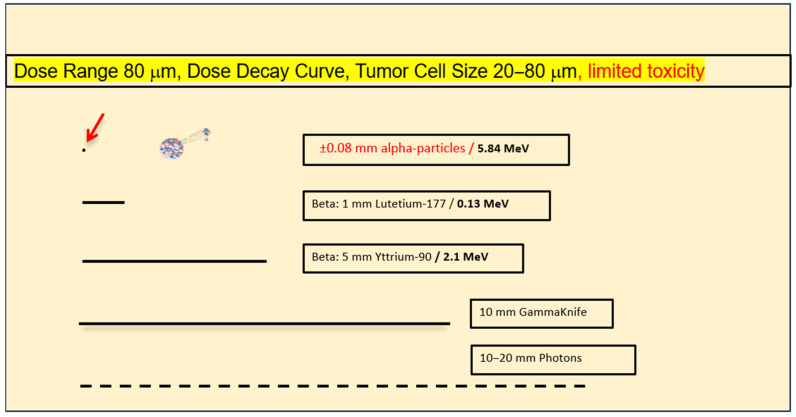
Dose decay curve from different radiation sources. Photon beams and gamma rays from Cobalt sources have a flat decay curve between 1 and 2 cm. Beta emitters range between 1 and 5 mm and alpha emitters range between 0.05 and 0.08 mm, which is 200× smaller than photon beams. The steepness of the decay curve defines toxicity.

**Figure 3 ijms-24-15701-f003:**
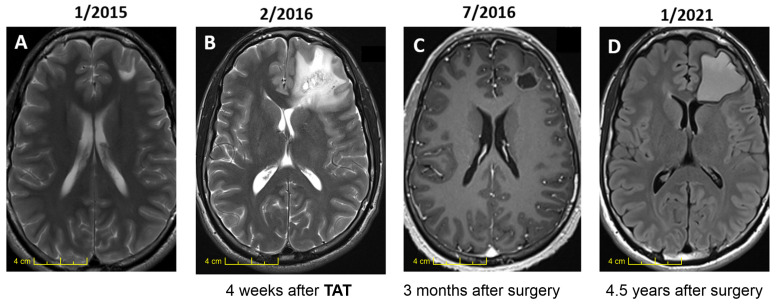
TAT demarcates the true initial tumor volume. In this 24 year old male patient, a small left frontal lesion was detected using MRI, following a focal seizure (image (**A**)). The biopsy disclosed a diffuse invasive grade 2 astrocytoma. Thus, 2 GBq [^213^Bi]Bi-DOTA-substance P was locally injected into the extracellular tumor matrix via a port–catheter system, inducing a strong perifocal edema reaction (image (**B**)). Two months later, the radionecrotic lesion was resected, creating a resectioned cavity of about 5 mL (image (**C**)). Over the next 4 years, the gradual resorption of the tumor margins manifested in a volume of about 50 mL (image (**D**)), at which point, it came to a halt. The true initial tumor volume is likely to have been 50 mL and not 2 mL, as shown on the initial MRI (image (**A**)). No neurological deficits ensued; KPS was 100.

**Figure 4 ijms-24-15701-f004:**
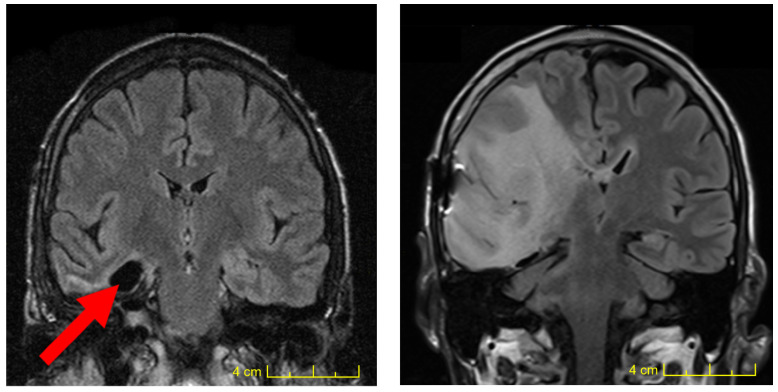
Massive tumor cell infiltration of normal brain tissue within nine years. In a typical case of small diffuse invasive astrocytoma, the visible tumor was resected in the right temporal lobe in a 36-year-old male patient. The red arrow denotes the initial resection cavity. Despite radio-chemotherapy, the lesion gradually expanded to a large mass that impaired proper neurological functioning and led to severe personality changes.

**Table 1 ijms-24-15701-t001:** Clinical data on low-grade glioma patients treated with TAT.

Case	Age&Year Dx	Histology/	Genetics	Pre-/Post-a	Activity/	Karnofsky	PS after	QALY
#	/Gender	Location		therapies	nuclide(cycle)	Performance	TAT /OS	
First-line TAT for LGG								
1	43(2000)m	oligo 2/pR	ND	S&Y-90SP/CT	2 GBq Bi-213(1)	90	288+/286+	23
2	33(2007)f	astro 2/fR	ND	none/S	2 GBq Bi-213(1)	100	192+/194+	16
3	39(2008)m	astro 2/oR	ND	none/S	2 GBq Bi-213(1)	100	180+/182+	15
4	64(2011)m	astro 2/centralR	IDH mut, 1p/19q wt	S/S	2 GBq Bi-213(1)	90	132+/150+	10
5	25(2011)m	astro 2/tL	IDH-1-R132H, ATRX mut	S/S	35 MBq Ac-225(2)	80	48+/144+	3.2
6	31(2013)f	astro 2/tL	IDH-1 mut, 1p/19qwt	S&RT/S	2 GBq Bi-213(1)	90	52+/120+	4
7	24(2015)m	astro 2/fL	IDH2 Exon4 R172M	none/S	2 GBq Bi-213(1)	100	96+/100+	8
8	32(2018)m	astro 2/fR	IDH-1 R132H, ATRX mut	S/none	20 MBq Ac-225(1)	100	22+/66+	1.8
9	30(2019)m	astro II/tL	IDH R132H, ATRX mut	S/none	17 MBq Ac-225(2)	100	18+/54+	1.5
Second-line TAT for recurrent OG2 after Y-90 SP								
10	SK43(2003)m	oligo 2/pR	ND	S&Y-90SP	2.5 GBq Bi-213(3)	90	48/224	3.6
11	BW31(2003)f	oligo 2/pL	ND	S&Y-90SP/CT	2 GBq Bi-213(1)	70	64/186	3.7

Abbreviations: “oligo 2” denotes oligodendroglioma WHO grade 2, “astro II” diffuse invasive astrocytoma WHO grade 2, “f” frontal, “t” temporal, “o” occipital, “L” left hemisphere, “R” right hemisphere, “ND” not done, “S” surgery, “CT” chemotherapy, “SP” substance P, “RT” external photon beam radiotherapy, “PS” progression-free survival, “OS” overall survival time, QALY see methods.

## Data Availability

Additional data connected to this study can be requested either by the corresponding author or by the co-authors.

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
