# Peer review of "Long-Term Tumor Control Following Targeted Alpha Therapy (TAT) of Low-Grade Gliomas (LGGs): A New Treatment Paradigm?"

_ijms, 2023, doi:10.3390/ijms242115701_

Round 1

Reviewer 1 Report

In this manuscript,

Krolicki et al. present 11 cases distributed between 8 astrocytoma and 3 oligodendroglioma to whom targeted alfa therapy was administered in order to assess the risk of neurotoxicity and the effect of the therapy itself.

The paper is of undoubtedly interest and the experiments along with the statistical analysis are well conducted.

I have only few minor points that might improve the quality of the manuscript and make it easier to be understood also by a non-specialized audience.

1)      I strongly recommend to create a summary table with the main information for all the patients (grade of astrocytoma, age of TAT delivery, additional therapy, degree of margin removal ecc.). It would be extremely helpful for the reader.

2)      Indeed in the manuscript the authors keep mentioning a table 1 but I cannot see it in the whole manuscript. Was it supposed to be in the supplementary material?

3)      Is it possible to stress more what are the median survival without TAT? I understand the pool of patients and the variability maybe too high for a Kaplan curve but still it would be good to state clearly how much is the improvement based on published survival rate.  

4)      Line 198: the chapter has been repeated twice. Remove the repetition.

5)      Line 46: rephrase the sentence

Author Response

Point-by-point response to  the reviewers’ comments

Dear Editor,

Thank you for allowing us to revise our paper. We would like to thank both reviewer 1 and reviewer 2 for their insightful commentary and acknowledge their understanding of the content of our submitted manuscript.The constructive advice of the peer reviewers has substantially improved our paper. Attached are our detailed responses to their comments.

All changes in the manuscript have been marked.

On the behalf of the authors,

Yours sincerely,

Prof. Dr. Adrian Merlo

Response to reviewer 1

Krolicki et al. present 11 cases distributed between 8 astrocytoma and 3 oligodendroglioma to whom targeted alfa therapy was administered in order to assess the risk of neurotoxicity and the effect of the therapy itself.

The paper is of undoubtedly interest and the experiments along with the statistical analysis are well conducted.

I have only few minor points that might improve the quality of the manuscript and make it easier to be understood also by a non-specialized audience.

  • We thank the reviewer for the understanding and highlighting the importance of the subject. We are also grateful for the precious feedback.
  • I strongly recommend to create a summary table with the main information for all the patients (grade of astrocytoma, age of TAT delivery, additional therapy, degree of margin removal ecc.). It would be extremely helpful for the reader.
  • We thank the reviewer for this recommendation. The Table has been submitted but not uploaded into to text for review. This will be corrected.
  • Indeed in the manuscript the authors keep mentioning a table 1 but I cannot see it in the whole manuscript. Was it supposed to be in the supplementary material?
  • Please see comment above, Table 1 was not uploaded during editing.
  • Is it possible to stress more what are the median survival without TAT? I understand the pool of patients and the variability maybe too high for a Kaplan curve but still it would be good to state clearly how much is the improvement based on published survival rate.
  • We thank the reviewer for pointing this out. There has been done a full statistical analysis by a mathematician using a method which is suitable to interpret low case numbers in orphan disease studies, so called Bayesian approach. In this analysis, the Kaplan Meier curve of a large NIH database with more than 2000 cases has been used as basis and the 8 astrocytoma 2 cases have been plotted unto this curve. It cannot produce a p-value, but a useful statistical range that underscores the high probability that the long survival times are a true TAT effect. We will add a shortened summary of that analysis to the publication in the result and discussion session and are ready to share parts of the report with the readers (it is part of a regulatory communication).

4)     Line 198: the chapter has been repeated twice. Remove the repetition.

  • Thank you for this correction, it has been done.
  • Line 46: rephrase the sentence
  • We thank the reviewer for this remark, the sentence has been rephrased..

Reviewer 2 Report

In the manuscript by Krolicki et al., the authors describe a small cohort of patients with infiltrative low-grade glioma who were treated with experimental targeted alpha therapy (TAT). They demonstrate good long-term outcomes in their cohort of patients and a favorable side effect profile. While of potential interest, there are several aspects of the paper that need to be addressed:

- The terminology used for brain tumor diagnoses is out of date. Based on current WHO guidelines (2021 5th Edition, classification of CNS tumors), Arabic numbers (1,2,3,4) should be used instead of Roman numerals (I, II, III, IV). Similarly, "diffuse invasive astrocytoma" is not a recognized diagnostic entity. Finally, the manner of diagnosis should be explicitly stated. Were IDH1/IDH2 and 1p/19q status specifically determined in all cases? For all cases of IDH mutant astrocytoma, was CDKN2A status specifically tested? 

- Table 1 is not present in the manuscript and not available for this reviewer. Another way of representing the data for each individual patient would be to create a Swimmer's plot, incorporating treatment history, progression times, and survival. 

- Figures 1 and 2 do not contribute to the article in any meaningful way. 

- The quality of writing throughout the manuscript must be improved. The introduction and discussion are too long, often with unnecessary information or information that belongs in other sections of the manuscript. The manuscript needs to be more focused and succinct. As discussed above, the Methods are incomplete, and Table 1 is not present for review. Finally, the manuscript requires extensive English language editing, which the authors might consider utilizing a professional English language editing service. 

Other points:

- On lines 63 and 64, cell counts are described but should likely include superscript font (i.e., 1011 tumor cells should be written with superscript to denote 10^11 cells). 

- On lines 328-329 of the discussion, it is unclear how Figure 4 related to the burden of tumor "on the patient, family and the professional work environment". 

Discussed above.

Author Response

Reply to reviewer 2

In the manuscript by Krolicki et al., the authors describe a small cohort of patients with infiltrative low-grade glioma who were treated with experimental targeted alpha therapy (TAT). They demonstrate good long-term outcomes in their cohort of patients and a favorable side effect profile. While of potential interest, there are several aspects of the paper that need to be addressed:

- The terminology used for brain tumor diagnoses is out of date. Based on current WHO guidelines (2021 5th Edition, classification of CNS tumors), Arabic numbers (1,2,3,4) should be used instead of Roman numerals (I, II, III, IV). Similarly, "diffuse invasive astrocytoma" is not a recognized diagnostic entity. Finally, the manner of diagnosis should be explicitly stated. Were IDH1/IDH2 and 1p/19q status specifically determined in all cases? For all cases of IDH mutant astrocytoma, was CDKN2A status specifically tested? 

  • We thank the reviewer for this remark. The terminology has been adapted in the text, replacing the Roman numerals with Arabic numbers. The term “diffuse invasive astrocytoma” has been replaced by “low-grade astrocytoma”. The genetic analyses are shown in Table 1 which has not been uploaded by editing the manuscript. This will be corrected. In the more recent cases, molecular analysis has been done, however, not so in the older cases diagnosed before the introduction of molecular pathology into glioma diagnostics.

- Table 1 is not present in the manuscript and not available for this reviewer. Another way of representing the data for each individual patient would be to create a Swimmer's plot, incorporating treatment history, progression times, and survival. 

  • Please see comment above (Table 1 has not been uploaded. This will be corrected).

- Figures 1 and 2 do not contribute to the article in any meaningful way. 

  • We thank the reviewer for pointing this out. These 2 figures are not useful for a clinician, indeed. However, the paper is addressed at a wider audience consisting of radiochemists, nuclear physicists, peptide and anorganic chemists that are not familiar with the clinical presentation of this orphan disease. We think that including the figures is justified since they have especially been created to facilitate comprehension of those non-medical specialists involved in this interdisciplinary research.

- The quality of writing throughout the manuscript must be improved. The introduction and discussion are too long, often with unnecessary information or information that belongs in other sections of the manuscript. The manuscript needs to be more focused and succinct. As discussed above, the Methods are incomplete, and Table 1 is not present for review. Finally, the manuscript requires extensive English language editing, which the authors might consider utilizing a professional English language editing service. 

  • We thank the reviewer for this advice. The professional English language service of the journal is being used for final editing. Since this is the first comprehensive report on long-term clinical data in this orphan indication using the mostly unknown local TAT approach, thorough explanations and citations of previous underlying work has to be presented in order to allow an understanding of the complex procedure not only to MDs, but also to scientists of fields related to this type of interdisciplinary research. Again, Table 1 has not been uploaded during editing of the manuscript, and will, of course, be added to the final manuscript.

Other points:

- On lines 63 and 64, cell counts are described but should likely include superscript font (i.e., 1011 tumor cells should be written with superscript to denote 10^11 cells). 

  • This has been corrected.

- On lines 328-329 of the discussion, it is unclear how Figure 4 related to the burden of tumor "on the patient, family and the professional work environment". 

  • We thank the reviewer for this point that needs clarification. Figure 4 has been introduced further up in the manuscript. The link to this figure has been removed as requested by the reviewer.

Round 2

Reviewer 1 Report

The Authors have done everything I asked for  and the manuscript has largely improved. 

I consequently recommend this manuscript for publication. 

Best,

Reviewer 2 Report

The authors have generally addressed my concerns with the article, though we disagree on some points. Regarding point 1, the term "low grade infiltrating astrocytoma", as "low grade astrocytoma" can also refer to pilocytic astrocytoma and other benign entities.

N/A